# Assessing Tourists' Preferences and Willingness to Pay for Artificial Beach Park Development and Management: A Choice Experiment Method

**Qi Chen [1,2,\*] and Yun Zhang [1]**

1    School of Business, Ningbo University, Ningbo 315211, China
2    Marine Economic Research Center, Donghai Academy, Ningbo University, Ningbo 315211, China
\*    Correspondence: chenqi2@nbu.edu.cn; Tel.: +86-134-5619-0651

**Abstract:** Artificial beaches have made a significant contribution to the expansion of coastal tourism. Obtaining information on tourists' preferences for artificial beach tourism can help managers to better balance the relationship between the satisfaction of recreational needs and environmental protection. The Meishan Bay Beach Park in Ningbo, Zhejiang Province, China was used as the study site, and the tourists' preferences for the artificial beach park development and management were evaluated using the choice experiment method. The results revealed that tourists were generally more interested in improving the governance level of the existing landscape than in further expanding the scale of artificial beach development. Among all attributes, significantly reducing the amount of garbage was the most preferred attribute, with a willingness to pay of 39.75 CNY, while willingness to increase beach area was the lowest attribute. The result of the preference heterogeneity analysis showed that tourists with higher education levels were more willing to pay to obtain a better recreational experience, while local tourists were more concerned about reducing congestion. Moreover, we found a clear and relevant segmentation of tourists' choice behavior, with the strong sensitivity for raising the ticket price being driven by the smallest group of the sample.

**Keywords:** artificial beach park; beach recreation; beach management; choice experiment method; willingness to pay; preference heterogeneity

## 1. Introduction

Beaches are a valuable natural resource with many important functions, such as coastal protection, ecological services, and recreation [1–3]. However, due to the impact of global climate change and human activities, about 70% of the beaches in the world are subject to erosion [4–6]. Beach nourishment is the process of moving sand onto an eroded coast and has become the most common way of mitigating beach erosion [7–9]. Beach nourishment includes two types of beach restoration and man-made beaches [10,11]. With the increasing demand for coastal tourism, the tourism and recreational functions of man-made beaches have been widely developed. Currently, man-made beaches have combined the dual functions of coastal protection and recreation for tourism [10]. Sand, sun, and sea (3S) have always been the main factors considered by tourists when choosing coastal scenic spots [12,13]. Due to the scarcity of natural sandy coasts, an increasing number of coastal cities choose to build landscape beaches or bathing beaches through artificial beaches.

China's sandy coasts are mainly concentrated in the southern and northern coastal areas and are scarce in the central coastal region. Since the 1990s, a series of artificial beach construction projects have been implemented in the central coastal regions of China. As of August 2019, a total of 44 artificial beach construction projects have been completed or are underway in China, mainly in the central coastal provinces and cities such as Zhejiang, Fujian, Shanghai, and Jiangsu [14]. The construction of artificial beaches has made outstanding contributions to the continuous expansion of China's coastal tourism

industry. Statistics showed that the average annual growth rate of the added value of China's coastal tourism industry reached 16% from 2008 to 2019 [15]. However, the development of coastal tourism also poses a serious threat to the coastal environment. Seawater pollution and beach litter result in increasingly severe damages to the environment and natural habitats. The sustainable development of coastal tourism has attracted increasing attention [16,17]. Balancing the relationship between meeting recreational needs and protecting the environment has become a key issue to be solved by the government and scenic spot managers [18,19].

Understanding tourists' preferences in coastal tourism development and management are essential for optimizing marketing and management strategies of scenic spots [20]. The contingent valuation method (CVM) was a common method used in early research to assess visitors' willingness to pay (WTP) for conservation goals and tourism development in coastal management. However, the CVM is unable to assess the relative importance of various attributes or information for potential trade-offs in tourism resource development and environment conservation [21,22]. In contrast, the choice experiment method (CEM) has received increasing attention by providing combinations of attributes across scenarios to understand variations in tourists' preferences rather than simply assessing a single specific scenario [23]. Several studies have focused on the application of CEM to assess tourists' preferences for coastal tourism resource conservation. For example, Xu and He used Nansha Wetland Park as the study site to evaluate tourists' willingness to pay for mangrove coverage, biodiversity conservation, and seawater quality improvement [24]. David et al. assessed recreational divers' preferences for the abundance and size of reef fishes [25]. There are also some studies focusing on tourists' preferences for coastal tourism management attributes. There have been also some studies focusing on the use of CEM to assess tourists' preferences for coastal tourism management attributes. For example, Boaz and Maya assessed beach visitors' preferences for the crowdedness of the beach and the presence of infrastructure [26]. Salpage et al. assessed tourists' preferences for ecotourism development and biodiversity conservation in coastal wetlands management [27]. He measured tourists' WTP for management attributes such as leisure project development and beach maintenance.

Although there have been many CEM studies conducted in natural landscapes in the coastal zone, few studies focus on the tourists' preference for artificial beach scenic spots. Previous studies have shown that visitors preferred natural beaches with little or no development; thus, reducing human intervention in the natural landscape was considered essential [26]. However, artificial beaches do not belong to natural landscapes, and their size and shape can be designed and changed. Therefore, tourists' preferences for the artificial beach development and management may be different from natural coastal landscapes. In general, the size of the beach can directly affect the recreational experience of visitors [28]. Determining the size of the beach area is a central element in planning for artificial beach development. Therefore, it is necessary to include beach size in the choice set to examine visitor preferences for further expansion of artificial beach development. Furthermore, similar to natural beach scenic spots, artificial beaches also face management problems such as overcrowding and environmental pollution. Previous studies have shown that massive influxes of tourists to beaches (often beaches with deficiencies in management) have a hugely negative impact on the environmental quality of marine ecosystems as well as significantly reducing the tourist experience [29,30]. Several surveys on visitor perceptions have confirmed that visitors are generally highly concerned about the presence of litter in sand and vegetation, as well as the condition of litter facilities [31–33]. Reducing beach litter pollution was considered a priority for beach management [31]. Artificial beaches are more sensitive to environmental changes due to their limited self-purification capacity [34]. The water quality, beach quality, and vegetation in the artificial beach scenic area require regular maintenance, which greatly increases the operating costs of the scenic area [14]. Therefore, understanding the trade-off between tourism development and environmental conservation is essential for the sustainable management of artificial

beaches. Altogether, it is necessary to specifically evaluate tourists' preferences for artificial beach park development and management, so as to provide a reference for the government and scenic spot managers to formulate planning and optimize management decisions.

This study ran a choice experiment in Meishan Bay Beach Park, Ningbo, Zhejiang, China, among 396 visitors, who were asked to choose from several options associated with various development and management strategies. We used the mixed logit model and latent class model to assess tourists' preference heterogeneity and estimate tourists' WTP for artificial beach park development and management attributes, which can provide a reference for scenic spot managers to adjust charging plans and optimize marketing strategies.

## 2. Materials

Meishan Bay Beach Park is located on the east coast of Beilun, Ningbo, Zhejiang Province, China. It is located on the south side of the Meishan waterway, adjacent to Yangsha Mountain (Figure 1). The park had its test run in July 2018. With a coastline of 1.88 kilometers and a crescent-shaped beach of 320,000 square meters, it is the largest artificial beach in East China and has become a remarkable coastal tourist destination in Ningbo. The construction of the park was completed in three phases, namely, the dredging and remediation of the southern and northern sections of the Meishan waterway, the first-phase project of artificial beaches, and the greening of the landscape and ancillary facilities of artificial beaches. The whole project started in September 2013 and was completed in June 2019, with a total investment of approximately 400 million CNY. Due to the vast area, various recreational sections, such as public square areas, public recreation areas, water recreation areas, and beach volleyball areas, have been set up. Since its operation and opening, Meishan Bay Beach Park has received over 600,000 tourists annually, with several peaks in passenger traffic during the high season. The surge in recreation demand poses a severe challenge to the supervision of scenic spots. Crowding seriously affects the tourist experience, safety, and satisfaction. Moreover, the coastal pollution in scenic spots is becoming increasingly serious. Many kinds of garbage, such as plastic, foam, and glass, have caused serious damage to coastal beaches and sea environments. Based on this, Meishan Bay Beach Park, which is rich in coastal tourism resources but is also faced with pollution and other management issues, was selected as the study site in our study. This can help to provide a better understanding of the way in which tourists make trade-offs between development and management attributes, and provide guidance for artificial beach management.

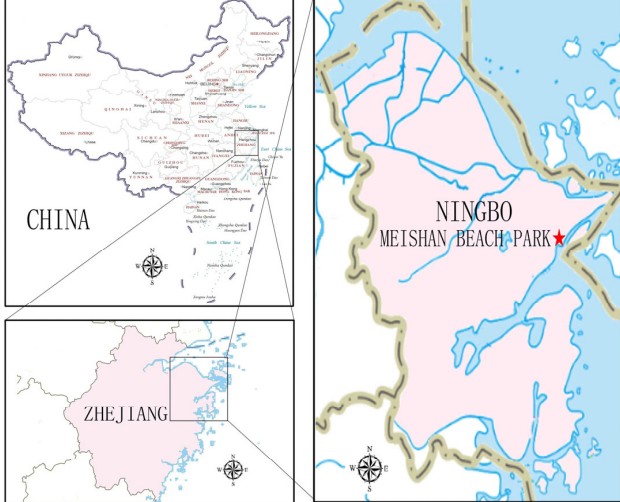

**Figure 1.** Location of the Meishan Beach Park. Notes: The maps of China, Zhejiang Province, and Ningbo City were retrieved from two websites and combined together by the authors (http://bzdt.ch.mnr.gov.cn; https://zhejiang.tianditu.gov.cn/standard, accessed on 15 April 2022).

### 3. Research Methods and Choice Experiment Design

*3.1. Research Methods*

The CEM is intended to obtain people's preference information by asking them to choose their preferred option in each choice set [35]. According to Lancaster's attribute value theory and McFadden's random utility theory in the behavioral analysis framework [36,37], the utility $U_{ij}$ that respondent *i* obtains from scenario *j* is given by:

$$U_{ij} = V_{ij} + \varepsilon_{ij} \tag{1}$$

where $V_{ij}$ represents the observable utility, which is explained by the attributes of the selected scenario *j*, and $\varepsilon_{ij}$ is a random term representing the unobservable part. According to the utility maximization principle, the respondent will choose the scenario that brings him or her the greatest utility, which means $U_{ij} > U_{ik}$ for any $k \neq j$ if respondent *i* chooses alternative *j*. The probability that respondent *i* chooses scenario *j* is calculated as follows:

$$P_{ij} = \Pr\big[(V_{ij} + \varepsilon_{ij}) > (V_{ik} + \varepsilon_{ik})\big] \, \forall k \neq j \tag{2}$$

The error term $\varepsilon_{ij}$ is assumed to be independent and identically distributed (IID) with a Gumbel distribution of Type I. Then, the probability of any particular scenario *j* being selected can be expressed by the multinomial logit model.

The assumption of the multinomial logit model means that tourists are regarded as a completely homogeneous group, so it cannot express the randomness of individual preferences, nor can it further analyze the heterogeneity of people's choice preferences. The mixed logit model, which assumes that a random variable obeys a specific probability distribution, reflects the randomness of personal preferences to analyze the heterogeneity of tourists' preferences, and then it overcomes the defects of the multinomial logit model [38]. In this case, the indirect utility function for visitor *i* to choose scenario *j* from the choice set can be expressed in linear form as follows:

$$V_{ij} = ASC + (\beta_k + \overline{\omega_k})X_{ij} + \delta_{ij} \tag{3}$$

where $X_{ij}$ represents the set of attributes of scenario *j* selected by respondent *i*, $\beta_k$ is the vector of random coefficients, $\overline{\omega_k}$ is the vector of random error term, and $\delta_{ij}$ is the random error term. *ASC* is an alternative specific constant used to explain the impact of unobservable attributes on choice outcomes and represents the baseline utility of respondents when choosing the status quo option. *ASC* is assigned a value of one when respondents choose the "status quo" option; otherwise, it is recoded as zero. Therefore, when the coefficient of the *ASC* item is negative, it indicates that tourists are more willing to pay for coastal tourism resource development and management. $\beta_j$ is the set of corresponding coefficients to be estimated.

Further considering the influence of individual heterogeneity factors on tourist preferences, the utility function can be extended to the following two specific forms:

$$V_{ij} = ASC + (\beta'_k + \overline{\omega_k})X_{ij} + \lambda_m \times ASC \times Z_{im} + \delta'_{ij} \tag{4}$$

$$V_{ij} = ASC + (\beta''_k + \overline{\omega_k})X_{ij} + \gamma_m \times X_{ij} \times Z_{im} + \delta''_{ij} \tag{5}$$

Equation (4) is developed to investigate the source of heterogeneity of tourists choosing a specific scenario, where $\delta'_{ij}$ is the random error term, $Z_{im}$ represents the individual demographic variables of tourists, $\lambda_m$ represents the coefficient vector of interaction terms for *ASC* and $Z_{im}$, and *m* represents the n-th individual demographic variables. Equation (5) is used to investigate the source of heterogeneity in tourist preferences for a specific attribute when choosing a scenario; $\delta''_{ij}$ is the random error term and $\gamma_m$ represents the interaction coefficients for $X_{ij}$ and $Z_{im}$.

To further analyze the preference heterogeneity among tourist groups, this paper also uses the latent class model to analyze the data. The latent class model divides a sample into

several latent classes based on observable attributes and unobservable variables. People in different classes have heterogeneous preferences while people in the same group have homogeneous preferences. The probability of tourist $i$ being classified into category $C$ and selecting scenario $j$ in the choice set is:

$$P_{ij|c} = \sum_c \frac{e^{\beta_{cj} X_{ij}}}{\sum_k e^{\beta_{cj} X_{ik}}} Q_{ic} \tag{6}$$

where $\beta_{cj}$ is the parameter vector corresponding to the tourist group of category $C$ and $Q_{ic}$ is the probability that tourist $i$ is classified into category $C$, which can be expressed as:

$$Q_{ic} = \frac{e^{\tau_c Z_i}}{\sum_c e^{\tau_c Z_i}} \tag{7}$$

Here, $Z_i$ represents the individual demographic variables of tourists classified into the category and $\tau_c$ is the parameter vector corresponding to the individual demographic variables in category $C$ [38].

On the basis of the above model, tourists' willingness to pay (*WTP*) for each attribute can be estimated according to the parameter estimation results using Equation (8), that is, the marginal value of the attribute.

$$WTP_k = -\frac{\beta_k}{\beta_p} \tag{8}$$

In this equation, $\beta_k$ represents the coefficient of the $k$-th attribute variable, and $\beta_p$ represents the coefficient of the monetary attribute variable (ticket price in this paper) in the scenario.

### 3.2. Design of the Choice Experiment

Determining the attributes and their levels and then constructing the choice sets are key to designing a choice experiment. To obtain more comprehensive information on tourist preferences for artificial beach park development and management, we divided the scenario attributes of the choice experiment into three types: development attribute, management attributes, and price attribute. On the basis of a pretest of tourists, scenic spot managers and experts, the literature review, and scenic area information selecting, it was finally determined that the beach area was the development attribute considered in our study; the seawater clarity, the amount of garbage, and the degree of congestion were chosen as the main management attributes of the study area. The ticket price was adopted as the price attribute. Coastal tourism has obvious seasonal characteristics and summer is the peak season. The description of the attribute level in this study was based on the state during the peak season. The attributes, levels of attributes, and the description of alternative levels are shown in Table 1.

The first attribute refers to an increase in the area of beach for the assessment of tourists' preference for expanding the scale of artificial beach development. Beaches, sea water, and sunshine are always the main factors considered by tourists when choosing coastal scenic spots [12,13]. At the beginning of its construction, Ningbo Meishan Bay Beach Park used a large crescent-shaped artificial beach as the core symbol to attract tourists. The first attribute level represents an increase in the beach area, and the second attribute level represents keeping the beach area unchanged (status quo).

The second attribute is seawater clarity. In addition to beaches, coastal tourism is largely dependent on good water quality [20]. Because Meishan Bay is a semi-closed bay, the seawater exchange capacity is weak, and the self-purification capacity is insufficient, which can easily lead to pollution problems. Some measures have been taken to improve water quality, such as setting up floating beds on the water, using shellfish and algae to absorb nutrients, and installing hydrodynamic devices to improve the fluidity of the water

body. Three levels of clarity, i.e., high, good, and basic, were finally defined after consulting experts, with the basic current level as a reference.

**Table 1.** Description of attributes and levels.

| Attribute | Levels | Description |
|---|---|---|
| Beach area | Increase | Widening the artificial beach |
| | Remain unchanged * | Keep the original area unchanged. |
| Seawater clarity | High | The visibility of sea water is about 3.5 m. |
| | Good | The visibility of sea water is about 2.5 m. |
| | Basic * | The visibility of sea water is about 1.5 m. |
| Litter Quantity | Little | The amount of litter is about 1 piece per 50 square meters. |
| | Less | The amount of litter is about 5 pieces per 50 square meters. |
| | Some * | The amount of litter is about 10 pieces per 50 square meters. |
| Congestion | Low | The number of tourists per 100 square meters is about 10. |
| | Moderate | The number of tourists per 100 square meters is about 20. |
| | High * | The number of tourists per 100 square meters is about 40. |
| Ticket Price (CNY/person) | 0 *, 20, 40, 60 | Single ticket price |

Note: * indicates the status quo. Beach tourism has obvious seasonality. The status quo of attributes and their improvement levels in this study were set based on peak season status.

The two attributes of litter quantity and congestion are not only important factors affecting tourists' recreational experience but are also practical problems that scenic spot managers need to solve. Most of the coastal beaches in Zhejiang Province are offshore island beaches. There are few on-shore beaches. The establishment of Meishan Bay Beach Park fills the gap well. Since the opening of the park, it has attracted a large number of tourists. However, it has also brought challenges to the management of the scenic spot. The results of the pretests showed that overcrowding during the peak tourist season in recreational areas is still complained about by tourists despite the large area of Meishan Bay Beach Park. At the same time, the surge in the number of tourists further increased the difficulty of controlling garbage disposal in scenic spots, and garbage in some core recreational areas cannot be treated in time. Combined with presurvey and expert opinions, three levels were designed for both the amount of garbage and the congestion attribute: little, less, and some for the former; low, moderate, and high for the latter.

The price attribute is used to reflect the fees that tourists are willing to pay to improve the attribute's level, which is measured by ticket price in this study. Meishan Bay Beach Park has been open to tourists free of charge since the trial operation in July 2018. However, the continuous surge in the number of tourists has brought increasing pressure on the management and operation of the scenic spot. Therefore, the scenic spot management has already considered formulating a charging plan. Taking into account the sensitivity of tourists to ticket prices, the standard fees of similar scenic spots, and the interview and investigation of scenic spot management, we set the ticket prices as 0 CNY, 20 CNY, 40 CNY and 60 CNY.

The choice sets were designed using the software package Ngene1.1.1. Based on the identified attributes and their corresponding levels, an orthogonal design was used to obtain choice tasks. Finally, 18 choice sets were included in this choice experiment. Each choice set contains three options: Options A and B are solutions to improve the status quo, and at least one attribute level in the scenario is improved; Option C is the status quo option, without any change in attribute levels. Table 2 is an example of the choice sets. If all 18 choice sets were included in a questionnaire, it would be too burdensome

for the respondents to make reasonable decisions. Therefore, each respondent was asked to randomly respond to a block of six choice tasks. The only difference between the three versions of the questionnaires was the choice set. Multi-version questionnaires can effectively reduce the complexity of tasks so that respondents have enough energy to fully consider the options, thus ensuring the stability of preference information [39,40]. After several rounds of pretests, a draft questionnaire was finalized. The questionnaire ultimately adopted mainly consists of three parts. The first part describes the background and the purpose of the investigation and includes an explanation of the attributes and their levels. The second part is the core of the questionnaire, which requires respondents to choose their preferred scenario from six choice sets in turn. The third part is a survey of respondents' socio-economic and demographic information.

**Table 2.** An example of choice set.

| Attribute | Option A | Option B | Option C (Status Quo) |
|---|---|---|---|
| Beach area  | Increase Widen the artificial beach | Remain unchanged Keep the original area unchanged | Remain unchanged Keep the original area unchanged |
| Seawater clarity  | Good The seawater visibility is about 2.5 m | High The seawater visibility is about 3.5 m | Basic The seawater visibility is about 1.5 m |
| Litter quantity  | Less The amount of litter is about 5 pieces/50 m$^2$ | Little The amount of litter is about 1 pieces/50 m$^2$ | Some The amount of litter is about 10 pieces/50 m$^2$ |
| Congestion  | High The number of tourists per 100 m$^2$ is about 40 | Moderate The number of tourists per 100 m$^2$ is about 20 | High The number of tourists per 100 m$^2$ is about 40 |
| Ticket Price  | CNY 40 | CNY 60 | CNY 0 |
| My option | ☐ Option A | ☐ Option B | ☐ Option C |

*3.3. Sample Characteristics*

Undergraduate and postgraduate students majoring in economics at the Business School of Ningbo University formed the investigation team for this study. Before conducting the survey, members of the team were trained by the authors. We explained the new programs, attributes, attribute levels, choice sets, and other relevant issues to ensure that the investigators fully understood the questionnaire. The preliminary survey was carried out in July 2019, and the survey subjects included tourists, managers, and experts. The purpose of the pre-survey for tourists was to assess whether visitors can complete the questionnaire accurately and efficiently, with a view to understanding whether the number of questions set in the questionnaire and the formulation of the questions were reasonable. The main purpose of the pre-survey for managers and experts was to obtain their general evaluation of and suggestions for the questionnaire to ensure the scientific and rational design of the attributes and their levels. On the basis of the presurvey, we

modified and improved the questionnaire to obtain the final questionnaire. The formal survey was conducted from September to October 2019. The team randomly distributed questionnaires to tourists in a face-to-face manner in Meishan Bay Beach Park. A total of 441 questionnaires were distributed, of which 396 effective questionnaires could be used for subsequent analysis. The socio-economic and demographic characteristics of the 396 respondents are shown in Table 3. The proportion of males and females in the sample is relatively balanced. The average age of the respondents was around 30 years old, more than 80% of the respondents had an education level of high school and above, and the average annual income of the respondents was about 70,000 CNY. Approximately 79.29% of the respondents lived in Ningbo, indicating that the scenic spot was mainly dominated by the local tourist market.

**Table 3.** Demographic characteristics of the sample.

| Variable | Assignment | Mean Value | Standard Deviation |
|---|---|---|---|
| Gender | Male = 1, female = 0 | 0.437 | 0.496 |
| Age | Actual age/year | 30.356 | 10.520 |
| Education | Junior high school or below = 1, Senior High School = 2, Graduate = 3, Higher degree = 4 | 2.354 | 0.777 |
| Annual income | Individual annual income/10,000 CNY | 7.101 | 4.218 |
| Residence | Ningbo = 1, Non-Ningbo = 0 | 0.793 | 0.405 |

## 4. Results

### 4.1. Mixed Logit Model Estimation Results

The assumption of the mixed logit model is more reasonable than that of the multinomial logit model, and the fitting effect of the latter is usually better. Thus, this paper used the mixed logit model to analyze the data. Distribution assumptions about coefficients must be made to estimate these models. This paper drew on existing research [41,42] and assumed the ticket price to be a fixed parameter variable. Other variables were assumed to be random parameter variables. Halton draws 100 was used, and the estimation results obtained by Stata16 are shown in Table 4. It shows that *ASC* is statistically significant at the level of 1% with a negative coefficient. According to the initial setting of *ASC* (*ASC* was recoded as one when the current scenario was chosen), we found that tourists are willing to pay more to improve the current situation of artificial beach park development and management. Meanwhile, the standard deviation of *ASC* is statistically significant at the level of 1%, indicating that tourists have obvious heterogeneity in the choice of improvement scenarios. All the parameters associated with development and management attributes are positive and statistically significant at the 1% level, showing that the improvement of the beach area, seawater clarity, litter quantity, and congestion can significantly improve tourists' utility. For the estimates of the standard deviation, Litter quantity less, Litter quantity little, and Congestion moderate are significant. This indicates that there are significant differences in the preferences of different tourists for these three attribute variables, while the preferences for other variables are more homogeneous. The ticket price variable is statistically significant at the 1% level, and the coefficient is negative. This is in line with expectations that higher ticket prices reduce the utility of tourists.

**Table 4.** Mixed logit model estimation results.

| Variable | Model 1 | |
| --- | --- | --- |
| | Coefficients | Std. Errors |
| *ASC* | −3.499 *** | 0.354 |
| Beach area increase | 0.193 *** | 0.055 |
| Seawater clarity high | 0.425 *** | 0.078 |
| Seawater clarity good | 0.218 *** | 0.074 |
| Litter Quantity little | 0.802 *** | 0.090 |
| Litter Quantity less | 0.285 *** | 0.084 |
| Congestion low | 0.434 *** | 0.104 |
| Congestion moderate | 0.474 *** | 0.072 |
| Ticket Price | −0.020 *** | 0.002 |
| Std. Devs | | |
| *ASC* | 3.344 *** | 0.324 |
| Beach area increase | −0.180 | 0.169 |
| Seawater clarity high | 0.056 | 0.145 |
| Seawater clarity good | 0.120 | 0.182 |
| Litter Quantity little | 0.603 *** | 0.136 |
| Litter Quantity less | −0.480 *** | 0.171 |
| Congestion low | 0.371 | 0.401 |
| Congestion moderate | −0.420 *** | 0.143 |
| Obs | 7128 | |
| Log Likelihood | −1932.012 | |
| AIC | 3898.025 | |
| BIC | 4014.843 | |

Note: *** Statistically significant at the 1% levels.

To investigate the sources of heterogeneity in tourist preferences for various attributes of artificial beach park development and management, this paper incorporated individual tourist demographic variables into the models. The standard deviation estimation results of the four variables of *ASC*, Litter quantity little, Litter quantity less, and Congestion moderate in Model 1 were significant. These four variables were set as random parametric variables. Then, we generated the interaction between these four variables and the individual demographic variables. The interaction terms and other variables were set as fixed-parameter variables, and Models 2, 3, 4, and 5 were constructed. Table 5 represents the marginal utility coefficient estimates for the different attributes. Compared to Model 1, the Log Likelihood, Akaike information criterion (AIC), and Bayesian information criterion (BIC) of these models improved. The sign and significance of the coefficients of each attribute variable did not change significantly.

Model 2 includes the interaction between *ASC* and individual socioeconomic characteristics of tourists to examine the heterogeneity of tourists' preferences for the improved scenarios (select option A or B). The standard deviation of the *ASC* in Model 2 is still significant at the 1% level, but it has reduced from 3.344 to 2.920, indicating that the differences in individual characteristics of tourists explain part of the heterogeneity of tourists' preferences. Specifically, the interaction between education level and *ASC* is significant and the coefficient is negative, indicating that tourists with higher education levels prefer to improve the development and management level of artificial beach parks. The interactions between other individual characteristic variables and *ASC* are not significant.

**Table 5.** Results of mixed logit model with interaction terms.

| Variable | Model 2 | | Model 3 | | Model 4 | | Model 5 | |
|---|---|---|---|---|---|---|---|---|
| | Coefficients | Std. Errors | Coefficients | Std. Errors | Coefficients | Std. Errors | Coefficients | Std. Errors |
| *ASC* | −3.039 *** | 0.997 | −3.430 *** | 0.345 | −3.446 *** | 0.349 | −3.442 *** | 0.347 |
| Beach area [increase] | 0.199 *** | 0.055 | 0.188 *** | 0.054 | 0.180 *** | 0.055 | 0.191 *** | 0.054 |
| Seawater clarity [high] | 0.431 *** | 0.078 | 0.433 *** | 0.077 | 0.439 *** | 0.078 | 0.440 *** | 0.077 |
| Seawater clarity [good] | 0.220 *** | 0.074 | 0.220 *** | 0.073 | 0.229 *** | 0.073 | 0.221 *** | 0.073 |
| Litter quantity [little] | 0.795 *** | 0.087 | 0.881 *** | 0.298 | 0.790 *** | 0.085 | 0.793 *** | 0.086 |
| Litter quantity [less] | 0.286 *** | 0.084 | 0.285 *** | 0.083 | 0.176 | 0.293 | 0.298 *** | 0.083 |
| Congestion [low] | 0.427 *** | 0.101 | 0.416 *** | 0.100 | 0.418 *** | 0.100 | 0.414 *** | 0.100 |
| Congestion [moderate] | 0.479 *** | 0.073 | 0.475 *** | 0.071 | 0.478 *** | 0.071 | 0.0380 | 0.295 |
| Ticket price | −0.020 *** | 0.002 | −0.020 *** | 0.002 | −0.020 *** | 0.002 | −0.020 *** | 0.002 |
| Interaction | Interact with ASC | | Interact with Litter quantity [little] | | Interact with Litter quantity [less] | | Interact with Congestion [moderate] | |
| Gender | 0.241 | 0.430 | −0.022 | 0.132 | −0.177 | 0.134 | −0.129 | 0.123 |
| Age | 0.323 | 0.227 | −0.029 | 0.075 | −0.007 | 0.077 | −0.045 | 0.072 |
| Education | −0.421 ** | 0.203 | 0.054 | 0.065 | 0.004 | 0.065 | 0.078 | 0.061 |
| Income | 0.067 | 0.111 | −0.056 | 0.036 | 0.056 | 0.037 | 0.017 | 0.035 |
| Residence | −0.405 | 0.606 | −0.273 | 0.212 | 0.288 | 0.231 | 0.347 ** | 0.166 |
| Std. Devs | | | | | | | | |
| *ASC* | 2.920 *** | 0.276 | 3.154 *** | 0.298 | 3.164 *** | 0.296 | 3.147 *** | 0.297 |
| Litter quantity [little] | 0.624 *** | 0.130 | −0.553 *** | 0.129 | −0.563 *** | 0.129 | −0.577 *** | 0.128 |
| Litter quantity [less] | 0.562 *** | 0.142 | −0.528 *** | 0.141 | −0.511 *** | 0.143 | 0.537 *** | 0.139 |
| Congestion [moderate] | 0.504 *** | 0.133 | −0.414 *** | 0.140 | −0.423 *** | 0.139 | 0.408 *** | 0.144 |
| Obs | 7128 | | 7128 | | 7128 | | 7128 | |
| Log Likelihood | −1922.609 | | −1925.187 | | −1925.974 | | −1924.830 | |
| AIC | 3881.217 | | 3886.375 | | 3887.949 | | 3885.661 | |
| BIC | 4004.907 | | 4010.064 | | 4011.638 | | 4009.350 | |

Note: **, *** Statistically significant at the 5% and 1% levels.

The Litter quantity little, Litter quantity less, and Congestion moderate are included in interactions with individual socioeconomic characteristics as additional explanatory variables in Models 3, 4, and 5, respectively. The results of Model 3 and Model 4 show that the interactions between individual characteristics and litter quantity are not statistically significant, which means that the individual characteristics considered in these models are not the source of the heterogeneity of garbage quantity preferences. However, the interaction term between congestion and residence is significant and positive in Model 5, which indicates that local tourists living in Ningbo pay more attention to reducing the crowds of tourists.

*4.2. Latent Class Model Estimation Results*

The choice of artificial beach park development and management options by tourists can be motivated by different reasons and purposes. A latent class model was estimated to further explore the preferences of different categories of tourist. The first step to estimate a latent class model is to determine the number of classes. Table 6 shows that when the number of classes is two, the AIC is smallest and increases with the number of classes; BIC is smallest when the number of classes is four, but the difference with the class numbers of two and three is insignificant. Therefore, considering the above information, it is finally determined that tourists can be divided into two potential classes.

**Table 6.** Criteria for determining the optimal number of classes.

| Classes | LLF | AIC | BIC |
| --- | --- | --- | --- |
| 2 | −1935.4 | 3725.3 | 3920.4 |
| 3 | −1870.9 | 3762.9 | 3918.2 |
| 4 | −1842.4 | 3799.9 | 3915.3 |
| 5 | −1813.7 | 3908.7 | 3984.4 |

The results of the latent class model are shown in Table 7. Class 1 accounts for 84.7% of the sample, which shows a stronger preference for all types of attributes, and this group has a higher level of education compared to Class 2. This result reflects the fact that more educated tourists tend to choose the improved option, which is similar to the results obtained from Model 2. Higher price sensitivity is shown in Class 2, and an increase in the ticket price will significantly reduce the overall utility of this group. Comparing the preferences of specific attributes, we can see that Litter quantity little is the preferred option of both Class 1 and Class 2. However, increasing beach area and clearer seawater only increase the utility of people in Class 1 and do not seem to have any effects on those in Class 2.

*4.3. Analysis of Willingness to Pay*

According to Formula (8), combining the estimation results of the mixed logit models and latent class models, we can calculate the amount that tourists are willing to pay for each attribute improvement. There is little difference in willingness to pay (WTP) calculated according to the different mixed logit models (Model 1, Model 2, Model 3, Model 4, and Model 5). Considering that Model 2 examines the heterogeneity of tourists' choices and has the best fitting effect, the following analysis is mainly based on the calculation results of Model 2 and the latent class model, namely Model 6. Table 8 reports the marginal willingness to pay estimates.

According to the results of Model 2, tourists have a positive willingness to pay for the improvement of various attribute levels, but there are great differences in their preferences. Tourists have the highest preference for Litter quantity little, with their WTP reaching 39.75 CNY, which is followed by Congestion moderate, Seawater clarity high, and Congestion low. The WTPs for the latter three attributes are not greatly different, all being between 21 and 24 CNY. Tourists have the lowest WTP for Beach area increase, at only 9.95 CNY. Further comparison of the WTPs for the same attribute at different levels shows

that tourists' preferences for both attributes of seawater clarity and litter quantity increase with the improvement of the attribute status. Specifically, tourists are willing to pay 11 CNY to improve the seawater clarity from being basic to being good, and 10.55 CNY more to further improve the clarity to being high. If the level of the litter quantity was reduced to less, tourists are willing to pay an extra 14.3 CNY. If the level of the litter quantity was further reduced to little, the tourists' willingness to pay will increase by another 25.45 CNY. However, tourists' WTP does not increase with the improvement of the state of congestion. Table 8 shows that tourists' WTP for Congestion low is 2.60 CNY lower than that for Congestion moderate.

**Table 7.** Latent class model estimates.

| Variable | Model 6 | | | |
| | Class 1 | | Class 2 | |
| | Coefficients | Std. Errors | Coefficients | Std. Errors |
| --- | --- | --- | --- | --- |
| *ASC* | −2.649 *** | 0.188 | 0.050 | 0.408 |
| Beach area <sup>increase</sup> | 0.190 *** | 0.051 | −0.046 | 0.204 |
| Seawater clarity <sup>high</sup> | 0.353 *** | 0.074 | 0.817 *** | 0.286 |
| Seawater clarity <sup>good</sup> | 0.169 ** | 0.072 | 0.457 | 0.293 |
| Litter quantity <sup>little</sup> | 0.732 *** | 0.078 | 0.845 ** | 0.326 |
| Litter quantity <sup>less</sup> | 0.235 *** | 0.076 | 0.640 ** | 0.300 |
| Congestion <sup>low</sup> | 0.361 *** | 0.098 | 0.648 * | 0.340 |
| Congestion <sup>moderate</sup> | 0.428 *** | 0.065 | 0.497 * | 0.264 |
| Ticket price | −0.015 *** | 0.002 | −0.064 *** | 0.010 |
| Share | 0.847 | | 0.153 | |
| Class membership | | | | |
| Gender | −0.271 | 0.306 | 0.000 | |
| Age | −0.230 | 0.143 | 0.000 | |
| Education | 0.315 ** | 0.144 | 0.000 | |
| Income | −0.100 | 0.083 | 0.000 | |
| Residence | −0.040 | 0.317 | 0.000 | |
| Constant | 1.771 *** | 0.625 | 0.000 | |

Note: *, **, *** Statistically significant at the 10%, 5%, and 1% levels.

**Table 8.** Willingness to pay for natural and managed attributes.

| Variable | Model 2 | Model 6 | |
| | | Class 1 | Class 2 |
| --- | --- | --- | --- |
| Beach area <sup>increase</sup> | 9.95 | 12.67 | - |
| Seawater clarity <sup>high</sup> | 21.55 | 23.53 | 12.77 |
| Seawater clarity <sup>good</sup> | 11.00 | 11.27 | - |
| Litter quantity <sup>little</sup> | 39.75 | 48.80 | 13.20 |
| Litter quantity <sup>less</sup> | 14.30 | 15.67 | 10.00 |
| Congestion <sup>low</sup> | 21.35 | 24.47 | 10.13 |
| Congestion <sup>moderate</sup> | 23.95 | 28.53 | 7.77 |

According to Model 6, the WTPs of the two groups are obtained, and the results show that people in Class 1 are willing to pay more for each attribute than those in Class 2. For Class 1, tourists have the highest WTP for the attribute Litter quantity little, followed by Congestion moderate, Congestion low, and Seawater clarity high. The tourists have the lowest WTP for the attribute Beach area increase. The preference order for attributes for people in Class 1 is essentially similar to that of people in Class 2. Compared with Class 1, Class 2 is more sensitive to ticket prices, and lower WTP for all attributes are revealed in this group. Litter quantity little and Seawater clarity high are the most important attribute variables for Class 2. It is worth noting that Class 1 and Class 2 have different preferences

for congestion at different levels. Class 1 is willing to pay more for Congestion moderate than for Congestion low, which is the same as the result under the overall sample of Model 2, while Class 2 seems to have a higher willingness to pay for Congestion low.

## 5. Discussion

### 5.1. Tourists' Evaluation of the Development and Management Attributes

Information on tourists' preferences for the development and management of artificial beach parks is an important basis for managers to develop planning and marketing strategies. In this study, beach area was selected as a development attribute, while seawater clarity, amount of litter, and congestion were selected as management attributes. The results of the preference assessment showed that tourists' preference for increasing beach area was significantly lower than management attributes such as improved seawater quality, reduced amount of litter, and reduced crowding. Furthermore, the estimation results of the latent class model found that the increase in the beach area does not improve the utility of the price-sensitive tourist group. This result suggests that tourists prefer to improve the utilization and management of existing tourism resources from a "quality" perspective rather than expanding the scale of development from a "quantity" perspective. Among all attributes, tourists have the highest WTP for Litter quantity little, reflecting that beach cleanliness is the most important factor in attracting tourists. This result is consistent with other studies on coastal tourism preferences based on natural coastal landscapes [43–45]. In reality, litter management is essential in maintaining beach ecological services and ensuring visitor safety [46]. Tourist preferences for another management attribute, congestion, also deserve more attention. This paper has found that tourists prefer Congestion moderate over Congestion low, which is inconsistent with general expectations. It has been found that tourist crowding perceptions do not directly affect tourism satisfaction, and different impacts can be revealed in different contexts [47,48].

### 5.2. Sources of Heterogeneity of Tourist Preferences

The estimation results of the model including individual characteristics showed that tourists with higher education levels were more likely to choose improvement options. Similar conclusions have been drawn from previous studies on tourists' travel preferences. Groups with higher education levels usually have a greater awareness of risk perception and ecological environment protection, and they are willing to pay higher fees for a better recreational experience [49–51]. From the perspective of specific attributes, tourists' preferences for the amount of garbage and the degree of crowding have apparent heterogeneity. The place of residence of tourists can explain the preference heterogeneity in the level of congestion. Our findings show that local tourists place more importance on reducing congestion. A possible explanation for this is that local tourists, who are more conveniently located in terms of transportation, visit beach parks more frequently than nonlocal tourists. Thus, crowding in areas such as entrances and core attractions is more likely to affect their satisfaction. In addition, studies have shown that there are differences in the focus of local tourists and nonlocal tourists with regards scenic spots. Local tourists tend to view local tourist attractions from the perspective of urban functions, and they pay more attention to management efficiencies such as environmental governance and passenger flow control in scenic spots [52]. The results obtained from the latent class model indicate that tourists can be subdivided into two groups, with the majority group (Class 1) being willing to pay higher entrance prices to improve the artificial beach park development and management. In contrast, the other group, with a relatively lower education level (Class 2), is susceptible to pressure from entrance prices, and their WTP for each attribute is much lower than that of Class 1. This result is consistent with results provided in previous studies conducted in other contexts [53,54]. Price-sensitive tourists may be ubiquitous.

*5.3. Control of Scenic Capacity*

The continuous increase in the number of tourists poses a severe threat to the coastal ecosystem and affects the leisure experience [55,56]. Therefore, scenic capacity has been a central issue that needs much more attention in tourism development and management [57,58]. Previous studies have mainly explored scenic capacity in terms of ecological carrying capacity and tourist satisfaction. The number of tourists determined by the ecological carrying capacity is usually considered the maximum capacity of the scenic area. It can ensure the maximum utilization of tourism resources [59,60]. Studies based on a tourist satisfaction perspective have mainly emphasized the potential loss of tourist recreational utility and welfare due to overcrowding [61]. A survey with a five-point Likert scale showed that tourists are sensitive to beach overcrowding, and they are very consistent in their desire to "control the number of beach tourists to avoid overcrowding" [58]. Based on the CE, the results of this paper show that tourist preference for congestion is not at the lowest level but rather at moderate. Therefore, for the management of coastal beach parks, blindly reducing the number of tourists may not be the best choice. Instead, it is necessary to seek a balance between protecting the ecological environment and ensuring a recreational experience. In reality, adjusting ticket prices is usually a market measure for managers to control the number of tourists [62]. The analysis based on the latent class model in this paper shows that approximately 15% of tourists are sensitive to ticket prices, and this part of the group will be most affected in the event of an increase in ticket price.

## 6. Conclusions

This study was conducted with the goal of analyzing overall tourist preferences for artificial beach park management and exploring determinants of these preferences by applying a choice experiment. On this basis, we further measured tourists' WTPs for various attributes. These findings can provide useful references for improving the efficiency of coastal tourism resource management, balancing the relationship between tourism development and environmental protection, and ultimately promoting the sustainable development of artificial beaches.

The analysis results of the mixed logit model show that improvements in the state of both natural and managed properties can significantly increase the recreational utility of tourists. From the perspective of preference heterogeneity, tourists with higher education levels are more willing to choose improvement scenarios, while local tourists place more emphasis on reducing crowding. The latent class model was used to further explore the heterogeneity of tourists' preferences. The results show that the larger group of tourists pay more attention to the improvement of artificial beach park development and management and are willing to pay higher entrance fees for this purpose, while the other smaller group of tourists are extremely resistant to the increase in entrance fees and have no significant preference for several attributes, such as increased beach area and higher seawater clarity.

Measuring WTP can more clearly reveal tourist preferences. We found that tourists have the highest WTP for reducing the amount of litter to a low level, followed by reducing congestion to a moderate level, and improving seawater clarity to a high level. Tourists have the lowest WTP to increase the beach area. In general, tourists are more concerned about managed attributes than about natural attributes. The study also found that the WTP for most attributes increased as their level improved. An exception is that tourists' willingness to pay for congestion decreases as it improves from a medium level to a low level, which may be a problem that scenic area managers need to pay attention to when regulating the number of tourists in the future. In addition, based on the latent class model, we found that there was a large gap in the WTPs between the two groups, but both groups were most willing to pay to reduce the amount of litter. The above measurement results of WTP can provide a decision-making reference for scenic spot managers to formulate a reasonable ticket price and seek to meet the balance between tourists' recreational needs and ecological environmental protection.



**Author Contributions:** Conceptualization, Q.C.; Funding Acquisition, Q.C.; Methodology, Q.C.; Validation, Q.C. and Y.Z.; Writing—Original Draft, Q.C. and Y.Z.; Writing—Review and Editing, Q.C. All authors have read and agreed to the published version of the manuscript.

**Funding:** This work was supported by the Zhejiang Soft Science Research Program Project (2021C35063), the National Social Science Foundation of China (19CGL039) and the Yongjiang Social Science Young Talent Cultivation Project.

**Institutional Review Board Statement:** Not applicable.

**Informed Consent Statement:** Not applicable.

**Data Availability Statement:** The data presented in this study are available on request from the corresponding author. The data are not publicly available due to the privacy and ethical.

**Acknowledgments:** We are grateful for the comments of the anonymous reviewers, which greatly improved the quality of this paper.

**Conflicts of Interest:** The authors declare no conflict of interest.

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
