# Peer review of "Assessing Tourists’ Preferences and Willingness to Pay for Artificial Beach Park Development and Management: A Choice Experiment Method"

_sustainability, doi:10.3390/su15032547_

Round 1
Reviewer 1 Report
Regarding the title -- "Assessing tourists' preferences and willingness to pay for the artificial beach park development and management: A choice experiment method" -- you need to remove "the" before "artificial beach park" as using "the" connotes a specific park or you need to use the location: "Assessing tourists' preferences and willingness to pay for the artificial beach park development and management in Zhejiang Province, China: A choice experiment method."
In the abstract at the end it is surprising to read about farmers and beach development -- it seems out of place.
Aside from "artificial beach park" the key words are too vague.
This sentence (line 31-33) makes no sense: "Beach nourishment includes two types of beach restoration and man-made beaches, in which man-made beaches are re-sand on a coast with few or no beaches to build a new beach [10,11]."
Man-made beaches in Florida still fulfill both functions of recreation for tourism and coastal protection from storms and floods which can negatively impact tourist locations and ecosystems.
It would be beneficial to provide short explanations of some of your concepts as not all readers will be familiar with them. For example, on line 96 you should further explain this: "We use the mixed logit model and latent class model, which allow for heterogeneity in tourists' preferences." Or you can at least say "Further described later in this paper..."
It appears that you are not testing for whether tourists prefer man-made or natural beaches -- your question or attribute was about whether the beach area that is there should be expanded. Given that over 600,000 people a year go to this beach it is clearly popular so that does not seem to be a main question.
Your July 2019 survey included tourists, managers, and experts so why did you only focus on tourists in the final survey? There is no explanation for this.
Line 284, check your grammar: "Three versions of the questionnaires were equally distributed, and finally a total of 441 questionnaires were gave out."
Given that tourists seem much more concerned about litter and congestion -- which have nothing per se to do with the beach being artificial -- the article seems to be off the mark. The main issue here is not artificial or natural beaches but quality of experiences at beaches in general. The same issues are found along Florida's Gulf Coast -- tourists and residents are concerned with litter at the beaches, traffic to them, and overcrowding on the beaches. I would put these concepts forward as the main ones not artificial beaches which is what the title directs us towards. The focus on management is correct as that is what is required to address the concerns.
It seems that there should be a much greater literature review of overtourism on beaches (which has an extensive literature) and plastic pollution on beaches as these are two of the biggest findings of the tourists' concerns.
After reading the whole article I still cannot make sense of your final sentence of the abstract: "Moreover, we found a clear and relevant segmentation of farmers’ choice behavior, with the strong sensitivity for raising the ticket price being driven by the smaller group of the sample." Where are these farmers again??
Author Response
Response to the Reviewer 1
Dear Reviewer,
We thank you for your kind constructive comments and suggestions, which helped us to improve the manuscript considerably. We have tried our best to revise our manuscript according to the comments. Based on the comments and suggestions, we have the response below:
(1) Regarding the title -- "Assessing tourists' preferences and willingness to pay for the artificial beach park development and management: A choice experiment method" -- you need to remove "the" before "artificial beach park" as using "the" connotes a specific park or you need to use the location: "Assessing tourists' preferences and willingness to pay for the artificial beach park development and management in Zhejiang Province, China: A choice experiment method."
Response: Thank you for your careful review. We have removed "the" before "artificial beach park".
(2) In the abstract at the end it is surprising to read about farmers and beach development -- it seems out of place.
Response: Thank you for your careful review. The word "farmers" here should be "tourists". This is a clerical error in our writing.
(3) Aside from "artificial beach park" the key words are too vague.
Response: Thank you for your careful review. Based on the study, we modified the keywords to “Artificial beach park, Beach recreation, Beach management, Choice experiment method, Willing-ness to pay and Preference heterogeneity”
(4) This sentence (line 31-33) makes no sense: "Beach nourishment includes two types of beach restoration and man-made beaches, in which man-made beaches are re-sand on a coast with few or no beaches to build a new beach [10,11]."
Man-made beaches in Florida still fulfill both functions of recreation for tourism and coastal protection from storms and floods which can negatively impact tourist locations and ecosystems.
Response: Thank you for your careful review. We have revised the presentation about the artificial beach background. (Lines 32-36)
(5) It would be beneficial to provide short explanations of some of your concepts as not all readers will be familiar with them. For example, on line 96 you should further explain this: "We use the mixed logit model and latent class model, which allow for heterogeneity in tourists' preferences." Or you can at least say "Further described later in this paper..."
Response: Thank you for your careful review. The description of the research method is presented in Section 3. Lines 158-165 detail the advantages of the mixed logit model in analyzing preference heterogeneity. The introduction section generally does not require a detailed description of the methods. Therefore, taking into account the review comments, we have removed “which allow for heterogeneity in tourists' preferences” from the introduction section to avoid any doubt from the readers.
(6) It appears that you are not testing for whether tourists prefer man-made or natural beaches -- your question or attribute was about whether the beach area that is there should be expanded. Given that over 600,000 people a year go to this beach it is clearly popular so that does not seem to be a main question.
Response: Thank you for your careful review. The subject of our study is not whether visitors prefer natural or artificial beaches, but rather visitors' preferences for multiple attributes related to artificial beach development and management. Also, we are not just studying whether the beach area should be expanded. Beach size is only one of the four development and management attributes. We include four attributes: beach area, water clarity, congestion, and litter quantity in the choice set to compare the differences in tourists' preferences for different attributes.
(7) Your July 2019 survey included tourists, managers, and experts so why did you only focus on tourists in the final survey? There is no explanation for this.
Response: Thank you for your careful review. The purpose of the pre-survey was to test the reasonableness of the questions in the questionnaire in order to further improve it. In the pre-survey, we consulted managers and experts for suggestions on the questionnaire, which led to the refinement of the attributes and their level settings. In the final survey, since the object of our study is tourists' preference, we only need to conduct a questionnaire survey on tourists. Lines 291-298 in the paper are our interpretation of the problem.
(8) Line 284, check your grammar: "Three versions of the questionnaires were equally distributed, and finally a total of 441 questionnaires were gave out."
Response: Thank you for your careful review. We have changed the wording of this sentence (lines 300-301).
(9) Given that tourists seem much more concerned about litter and congestion -- which have nothing per se to do with the beach being artificial -- the article seems to be off the mark. The main issue here is not artificial or natural beaches but quality of experiences at beaches in general. The same issues are found along Florida's Gulf Coast -- tourists and residents are concerned with litter at the beaches, traffic to them, and overcrowding on the beaches. I would put these concepts forward as the main ones not artificial beaches which is what the title directs us towards. The focus on management is correct as that is what is required to address the concerns.
Response: Thank you for your careful review. We tried to respond to this comment from the following aspects:
First, the subject of this study is not a comparison of tourists' preferences for artificial and natural beaches; the object of our study and the issue of concern is tourists' preferences for the development and management of artificial beaches. Although there have been many tourists' preferences studies conducted in natural beaches, few studies focus on the tourists' preference for artificial beaches. Unlike natural beaches, the size and shape of artificial beaches can be designed and changed. Therefore, tourists' preferences for the artificial beach development and management may be different from natural coastal landscapes. Based on this, we designed choice experiments specifically for the development and management of artificial beaches to analyze tourists' preferences. We have added explanations about why artificial beaches were chosen as the study subject in lines 78-84.
Second, the higher preference for litter and congestion suggests that visitors are more concerned about management attributes than development attributes. This finding is the result of our analysis after including beach area, water quality, trash and congestion together in the choice set. However, we cannot dismiss the concern for beach area and water quality attributes in the development and management of artificial beaches in the preliminary choice experiment design. We have added some explanations in lines 436-440 of the paper.
Therefore, our research topic is tourists' preferences for various attributes of artificial beaches. We retain the term "artificial beach" in the title.
(10) It seems that there should be a much greater literature review of overtourism on beaches (which has an extensive literature) and plastic pollution on beaches as these are two of the biggest findings of the tourists' concerns.
Response: Thank you for your careful review. Based on your suggestions, we have added literature review of overtourism on beache and plastic pollution on beaches (Lines90-96).
We are very grateful to your comments for the manuscript. Your careful review has helped to make our study clearer and more comprehensive.
Sincerely.
Reviewer 2 Report
Dear researchers,
I enjoyed reading your paper. I think it is very interesting and well-written. You correctly identified a gap in the literature and found the right research methods to investigate the problem. The paper is coherent and easy to read, the hypotheses are clearly stated and the results are clearly presented. In my opinion, this research paper covers all aspects very well. There is not much I could add. The only thing is that the subject of artificial beaches is somewhat of a niche subject because, outside China, I don't think there are many such man-made beaches that are part of urban protected parks. I have only two minor observations;
1. Table 3 - I don't think calculating mean value is relevant in this context. In my opinion, the conservative way of using percentages would be better.
2. Most of the visitors seem to be locals. How ethical do you think it would be to impose restrictions on local population? I think it would have been interesting to see whether or not there were statistically significant differences between residents and tourists in the way they approached the problem.
All in all, excellent work, congratulations!
Author Response
Response to the Reviewer 2
Dear Reviewer,
We thank you for your kind constructive comments and suggestions, which helped us to improve the manuscript considerably. We have tried our best to revise our manuscript according to the comments. Based on the comments and suggestions, we have the response below:
(1) Table 3 - I don't think calculating mean value is relevant in this context. In my opinion, the conservative way of using percentages would be better.
Response: Thank you for your careful review. Since our variables contain not only dummy-assigned variables but also actual survey values such as age and income. Therefore, we did not use percentages for the statistics.
(2) Most of the visitors seem to be locals. How ethical do you think it would be to impose restrictions on local population? I think it would have been interesting to see whether or not there were statistically significant differences between residents and tourists in the way they approached the problem.
Response: Thank you for your careful review. Our survey was a random survey of visitors in the artificial beach park. The proportion of local and non-local visitors in the survey objectively reflects the source of visitors to the scenic area. Due to the relatively small sample size of non-local visitors, the statistical analysis for non-local visitors may have a large error. The comments you made are a very interesting direction for future research. In the future, we can try to expand the scope of the survey and conduct more in-depth research on the preferences of local and non-local tourists.
We are very grateful to your comments for the manuscript. Your careful review has helped to make our study clearer and more comprehensive.
Sincerely.
Round 2
Reviewer 1 Report
Thank you for addressing the comments and fixing the issues from the previous version.